# Breakthrough infection elicits hypermutated IGHV3-53/3-66 public antibodies with broad and potent neutralizing activity against SARS-CoV-2 variants including the emerging EG.5 lineages

Ling Li[1⊙], Xixian Chen[1,2⊙], Zuowei Wang[1], Yunjian Li[1], Chen Wang[1], Liwei Jiang[1]*, Teng Zuo[1]*

1 Laboratory of Immunoengineering, Institute of Health and Medical Technology, Hefei Institutes of Physical Science, Chinese Academy of Sciences, Hefei, People's Republic of China, 2 University of Science and Technology of China, Hefei, People's Republic of China

⊙ These authors contributed equally to this work.
* jlw0531@cmpt.ac.cn (LJ); zuot@cmpt.ac.cn (TZ)

**Data Availability Statement:** All relevant data are within the manuscript and its Supporting Information files.

## Abstract

The rapid emergence of SARS-CoV-2 variants of concern (VOCs) calls for efforts to study broadly neutralizing antibodies elicited by infection or vaccination so as to inform the development of vaccines and antibody therapeutics with broad protection. Here, we identified two convalescents of breakthrough infection with relatively high neutralizing titers against all tested viruses. Among 50 spike-specific monoclonal antibodies (mAbs) cloned from their B cells, the top 6 neutralizing mAbs (KXD01-06) belong to previously defined IGHV3-53/3-66 public antibodies. Although most antibodies in this class are dramatically escaped by VOCs, KXD01-06 all exhibit broad neutralizing capacity, particularly KXD01-03, which neutralize SARS-CoV-2 from prototype to the emerging EG.5.1 and FL.1.5.1. Deep mutational scanning reveals that KXD01-06 can be escaped by current and prospective variants with mutations on D420, Y421, L455, F456, N460, A475 and N487. Genetic and functional analysis further indicates that the extent of somatic hypermutation is critical for the breadth of KXD01-06 and other IGHV3-53/3-66 public antibodies. Overall, the prevalence of broadly neutralizing IGHV3-53/3-66 public antibodies in these two convalescents provides rationale for novel vaccines based on this class of antibodies. Meanwhile, KXD01-06 can be developed as candidates of therapeutics against SARS-CoV-2 through further affinity maturation.

## Author summary

The greatest challenge for the development of effective vaccines and antibody therapeutics against SARS-CoV-2 is the rapid emergence of variants. Broadly neutralizing antibodies (bnAbs) and vaccines that can elicit those antibodies are promising strategies to meet the challenge. Therefore, we focus on identification and characterization of bnAbs elicited by

**Funding:** This work was supported by the National Key Plan for Scientific Research and Development of China (L.J., T.Z., and L.L., Grant No. 2022YFC2305800, https://www.most.gov.cn/index.html) and the National Natural Science Foundation of China (T.Z., Grant No. 32200765, https://www.nsfc.gov.cn/publish/portal0/). The funders had no role in study design, data collection and analysis, decision to publish, or preparation of the manuscript.

infection or vaccination in this study. From two convalescents of breakthrough infection, we discovered a serial of bnAbs (KXD01-06), which are able to neutralize SARS-CoV-2 from prototype to XBB lineages, and even to the emerging EG.5. Notably, these bnAbs are prevalent in the two convalescents and their neutralization breadth is developed through extensive affinity maturation. Overall, our study is an important support for vaccine development aiming at eliciting those bnAbs. Moreover, those bnAbs have potential to be developed as candidates of therapeutics against SARS-CoV-2 through further optimization.

## Introduction

Since the outbreak of pandemic caused by SARS-CoV-2, numerous successes have been achieved in the development of vaccines and antibody therapeutics. Hundreds of vaccine candidates have been developed and 50 vaccines have been approved as of December 2022 (https://covid19.trackvaccines.org/). Meanwhile, tens of thousands of neutralizing monoclonal antibodies (mAbs) have been identified and 11 mAbs have successively received emergency use authorization within the first two years of the pandemic [1]. However, these successes have been gradually destroyed by the continuous emergence of SARS-CoV-2 variants. According to the transmissibility, virulence and immune escape of variants, the World Health Organization (WHO) has identified a succession of variants of concern (VOCs), including Alpha (B.1.1.7), Beta (B.1.351), Gamma (P.1), Delta (B.1.617.2) and Omicron (B.1.1.529). Omicron was first reported in South Africa in November 2021 and quickly became predominant worldwide [2]. To date, Omicron has developed hundreds of subvariants, which are classified into five main lineages including BA.1, BA.2, BA.3, BA.4, BA.5 and a serial of sublineages such as BA.1.1, BA.2.12.1, BA.2.75, BA.4.6, BF.7, BQ.1 and XBB [3]. Compared with SARS-CoV-2 prototype and earlier VOCs, Omicron lineages, particularly the emerging XBB and EG.5 sublineages, display most striking neutralization evasion from infection- and vaccination-elicited polyclonal antibodies and previously identified mAbs [4–10]. The persistent evolution of SARS-CoV-2 highlights the urgency to develop vaccines and antibody therapeutics with broad protection against current and future SARS-CoV-2 variants.

Spike (S) protein on virus surface mediates viral entry into host cells and is therefore the main target of vaccines and antibody therapeutics. Structurally, S protein is trimeric and each monomer consists of two functional subunits: S1 for engaging the receptor angiotensin converting enzyme 2 (ACE2) and S2 for driving fusion of viral and cellular membranes. S1 contains an amino-terminal (N-terminal) domain (NTD), a receptor-binding domain (RBD) and two carboxy-terminal (C-terminal) domains (CTD1 and CTD2). RBD further consists of a core structure and a receptor binding motif (RBM) that contacts with ACE2. S2 includes the N-terminal fusion peptide and its proximity region, heptad repeat 1 (HR1), central helix, stem helix, HR2, transmembrane region, and cytoplasmic tail. Although all the extracellular domains are susceptible to antibody binding, the majority of neutralizing antibodies target RBD with the rest recognizing NTD, S2, CTD or other epitopes [11,12].

A large-scale V gene usage analysis has revealed that RBD-specific antibodies are most frequently encoded by IGHV3-53 and the highly related IGHV3-66, and therefore these antibodies are termed as IGHV3-53/3-66 public antibodies [13]. Exemplified by CB6 (LY-CoV016, Etesevimab) and P2C-1F11 (BRII-196, Amubarvimab), IGHV3-53/3-66 public antibodies exhibit potent neutralizing activity against SARS-CoV-2 and provide effective protection against viral infection and disease progression in humans [1,14,15]. Nonetheless, most

antibodies in this family have been escaped by VOCs, which harbor mutations such as N417T/ K, N460K, E484K/A and F486V/S [4,16–19]. Although a few antibodies, such as COV11, R40-1G8, COVOX-222, P5-1C8, P5S-2B10, VacBB-551, BD55-1205 and BD56-1854, maintain broadly neutralizing activity across VOCs [8,19–23], they are extremely rare given the prevalence of IGHV3-53/3-66 antibodies.

Understanding the characteristics of broadly neutralizing antibodies (bnAbs) against SARS-CoV-2 will guide the rational design of vaccines and antibody therapeutics with broad protection. In this study, we identified two convalescents of breakthrough infection with relatively high neutralizing titers against all tested viruses including BQ and XBB lineages. Among 50 spike-specific monoclonal antibodies (mAbs) cloned from their B cells, the top 6 neutralizing mAbs (KXD01-06) belong to IGHV3-53/3-66 public antibodies and exhibit broad neutralizing capacity. Deep mutational scanning reveals that they target relatively conserved sites on RBD. Genetic and functional analysis indicates that the extent of somatic hypermutation is critical for their breadth. These findings provide rationale for the development of vaccines and antibody therapeutics based on IGHV3-53/3-66 public antibodies with broad and potent neutralizing activity.

## Results

### Identification of individuals with broad neutralizing activity

To characterize bnAbs elicited by infection or vaccination, we collected peripheral blood samples from 11 donors (Table 1). Except Donor 11 who was not vaccinated with any SARS-CoV-2 vaccine, all the other donors were vaccinated with either two doses of mRNA vaccine (Donor 1 and 2), or two doses of inactivated vaccine (Donor 3–6), or three doses of inactivated vaccine (Donor 7–10). During the wave of infection in December 2022, all donors except Donor 10 were infected with SARS-CoV-2, probably BA.5 or BF.7 variants as they were main circulating strains at that time [24].

We measured plasma binding antibody titers against prototype (wild-type, WT), BA.4/5 and XBB.1.5 spike by ELISA (Fig 1A and 1B). The antibody titers vary in a wide range with highest titers from Donor 2 and lowest titers from Donor 10 and 11. Moreover, antibody titers against WT, BA.4/5 and XBB.1.5 spike gradually decrease, which correlate with the number of mutations on these spikes. We further measured plasma neutralizing antibody titers against 19 pseudoviruses including WT, Delta, BA.1, BA.2, BA.3, BA.4/5, BA.2.75, BF.7, BQ.1, XBB,

**Table 1. Information of the donors and samples.**

| Donor | Gender | Age | Vaccine (1st dose) | | Vaccine (2nd dose) | | Vaccine (3rd dose) | | Infection time | Sampling time |
|---|---|---|---|---|---|---|---|---|---|---|
| | | | Time | Type | Time | Type | Time | Type | | |
| 1 | female | 36 | 2021/5/22 | mRNA | 2021/6/14 | mRNA | | | 2022/12/17 | 2023/2/10 |
| 2 | male | 35 | 2021/1/22 | mRNA | 2021/2/13 | mRNA | | | 2022/12/17 | 2023/2/10 |
| 3 | female | 29 | 2021/7/6 | inactivated | 2021/8/13 | inactivated | | | 2022/12/21 | 2023/2/10 |
| 4 | female | 28 | 2021/5/11 | inactivated | 2021/7/12 | inactivated | | | 2022/12/22 | 2023/2/10 |
| 5 | male | 25 | 2021/5/17 | inactivated | 2021/6/18 | inactivated | | | 2022/12/20 | 2023/2/10 |
| 6 | female | 26 | 2020/12/28 | inactivated | 2021/1/15 | inactivated | | | 2022/12/15 | 2023/2/10 |
| 7 | female | 26 | 2021/5/17 | inactivated | 2021/6/23 | inactivated | 2021/12/27 | inactivated | 2022/12/26 | 2023/2/10 |
| 8 | male | 28 | 2021/5/15 | inactivated | 2021/6/5 | inactivated | 2021/12/27 | inactivated | 2022/12/10 | 2023/2/10 |
| 9 | female | 31 | 2020/12/28 | inactivated | 2021/1/15 | inactivated | 2021/11/28 | inactivated | 2022/12/16 | 2023/2/10 |
| 10 | male | 28 | 2021/5/14 | inactivated | 2021/6/10 | inactivated | 2021/12/22 | inactivated | Not infected | 2023/2/10 |
| 11 | female | 31 | Not vaccinated | | | | | | 2022/12/21 | 2023/2/10 |

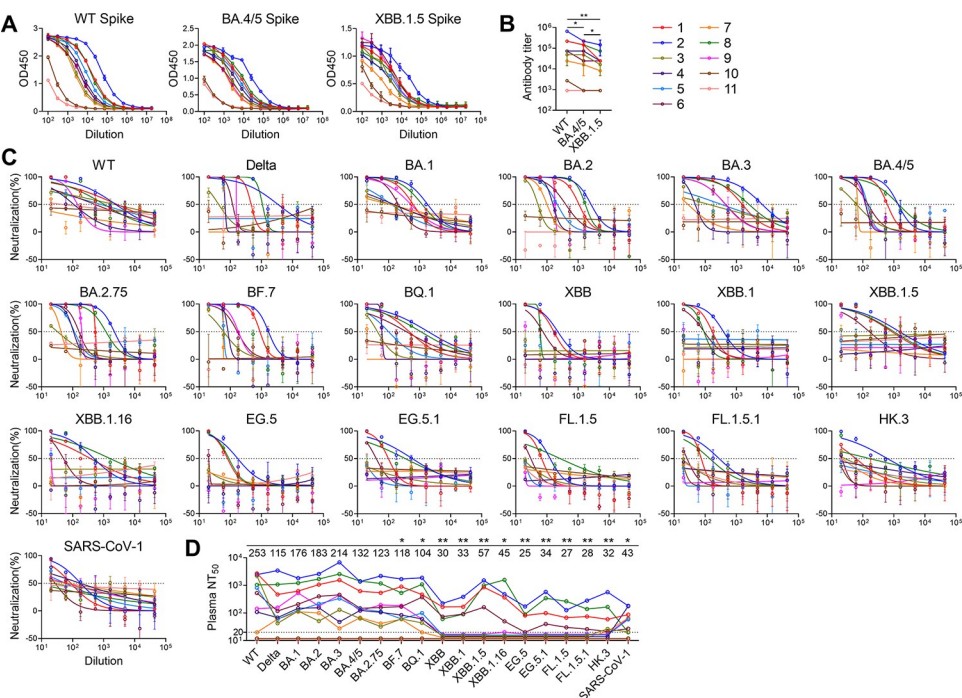

**Fig 1. Characterization of plasma from SARS-CoV-2 convalescents or vaccinees.** (A) Measurement of antibody titers against WT, BA.4/5 and XBB.1.5 spikes by ELISA. Data are represented as the mean ± SD. (B) Summary of antibody titers. Statistical analysis was performed by two-tailed Wilcoxon matched-pairs signed rank test. *P < 0.05, **P < 0.01. (C) Measurement of neutralizing antibody titers against a panel of pseudoviruses. Data are represented as the mean ± SD. (D) Summary of neutralizing antibody titers. For plasma with no neutralizing activity at 20-fold dilution, a number between 10–20 is given as titer to separate the curves. The numbers on top are geometric mean titers against the viruses. The titers against WT are compared with titers against other viruses. Statistical analysis was performed by two-tailed Wilcoxon matched-pairs signed rank test. *P < 0.05, **P < 0.01. All results are representatives of two independent experiments, in which duplicates are performed.

XBB.1, XBB.1.5, XBB.1.16, EG.5, EG.5.1, FL.1.5, FL.1.5.1, HK.3 and SARS-CoV-1 (Fig 1C and 1D). Donor 10 and 11 show little neutralizing activity, consistent with their low binding antibody titers. In contrast, Donor 1, 2, 6 and 8 exhibit broad neutralizing activity against all tested viruses. Donor 3, 4, 5, 7 and 9 have neutralizing titers against viruses before XBB lineages. The geometric mean titers against each virus further confirm that XBB lineages and variants with F456L and L455F are most resistant to neutralization, which is consistent with other studies [5–10].

## Isolation of mAbs from two convalescents

As Donor 1 and 2 have higher neutralizing titers against XBB and XBB.1 than other donors, we chose their blood cell samples to isolate mAbs. With WT spike as bait, we sorted single antigen-specific memory B cells and plasmablasts (S1A Fig). From those cells, we cloned 72 mAbs by single cell RT-PCR (Figs 2A, 2B, and S1B). By screening supernatant of 293T cells transfected with antibody-expressing vectors, we identified 50 mAbs positive for WT spike, among which 25 mAbs are also positive for WT RBD (Fig 2A and 2B). We further tested the neutralizing activity of spike-positive supernatant against WT pseudovirus. In sum, we identified 6 mAbs (KXD01-06) with potent neutralizing activity and 3 mAbs (KXD07-09) with moderate neutralizing activity (Fig 2B). Notably, KXD01-06 are all encoded by IGHV3-53/3-66, with either short CDRH3 ranging from 10 to 12 amino acids or long CDRH3 of 21 amino acids (IMGT numbering). Moreover, light chain V genes of these mAbs, including IGKV1-9,

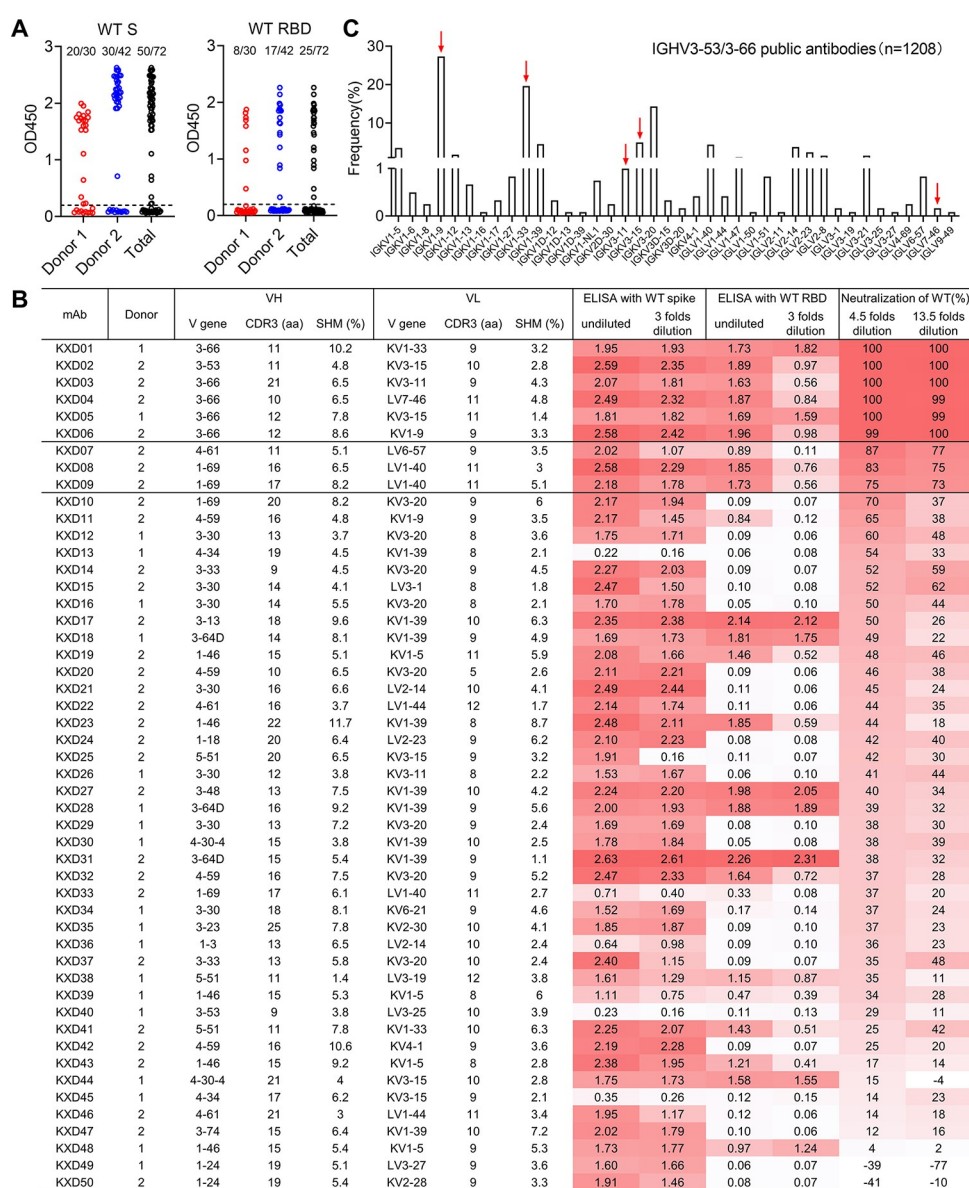

**Fig 2. Isolation and characterization of mAbs from Donor 1 and 2.** (A) Summary of mAb screening by ELISA. Undiluted supernatant from 293T cells transfected with antibody-expressing vectors was tested against WT spike and RBD. OD450 was measured and positive clones were identified with 0.2 as cut-off. (B) Summary of 50 spike positive mAbs. The mAbs are ordered based on the neutralizing activity of 293T supernatant. OD450s and neutralization percentages are color-coded, with darker red indicates higher values. (C) Frequency of light chain V gene usage among 1208 IGHV3-53/3-66 public antibodies. The red arrows indicate V genes used by KXD01-06.

IGKV1-33, IGKV3-11 and IGKV3-15, have been frequently reported to pair with IGHV3-53/ 3-66 (Fig 2C). These data suggest that a large portion of neutralizing antibodies from these two convalescents belong to previously defined IGHV3-53/3-66 public antibodies.

## Characterization of KXD01-09

To further characterize those neutralizing mAbs, we produced KXD01-09 together with COV2-2196, COV2-2130, P2C-1F11, P5-2H11, P5S-2B10, BD55-1205 and BD56-1854.

| mAbs | Blocking (%) | | | | RBD binding (EC50:µg/ml) | | | | | | Neutralization (IC50:µg/ml) | | | | | | | | | | | | | | | | | | |
| --- | --- | --- | --- | --- | --- | --- | --- | --- | --- | --- | --- | --- | --- | --- | --- | --- | --- | --- | --- | --- | --- | --- | --- | --- | --- | --- | --- | --- | --- |
| | ACE2-bio | CoV-2196-bio | CoV-2130-bio | P2C-1F11-bio | WT | BA.2.75 | BA.4 | BQ.1 | XBB.1.5 | EG.5.1 | WT | Delta | BA.1 | BA.2 | BA.3 | BA.4/5 | BA.2.75 | BF.7 | BQ.1 | XBB | XBB.1 | XBB.1.5 | XBB.1.16 | EG.5 | EG.5.1 | FL.1.5 | FL.1.5.1 | HK.3 | SARS-CoV-1 |
| KXD01 | 95 | 97 | 6 | 97 | 0.0025 | 0.0028 | 0.0022 | 0.0014 | 0.0010 | 0.0096 | 0.019 | 0.044 | 0.017 | 0.018 | 0.019 | 0.011 | 0.014 | 0.012 | 0.059 | 0.027 | 0.020 | 0.016 | 0.016 | 0.349 | 0.253 | 0.514 | 0.507 | >10 | >10 |
| KXD02 | 85 | 94 | 26 | 95 | 0.0042 | 0.0031 | 0.0037 | 0.0026 | 0.0027 | 0.0067 | 0.031 | 0.020 | 0.013 | 0.014 | 0.036 | 0.011 | 0.074 | 0.018 | 0.147 | 0.163 | 0.215 | 0.077 | 0.121 | 0.225 | 0.100 | 0.376 | 0.196 | 2.449 | >10 |
| KXD03 | 91 | 98 | 15 | 97 | 0.0032 | 0.0028 | 0.0028 | 0.0014 | 0.0015 | 0.0100 | 0.024 | 0.062 | 0.022 | 0.047 | 0.038 | 0.042 | 0.022 | 0.064 | 0.361 | 0.479 | 0.250 | 0.090 | 0.051 | 0.489 | 0.238 | 0.340 | 0.341 | 0.232 | >10 |
| KXD04 | 87 | 84 | 25 | 94 | 0.0060 | 0.0197 | 0.0132 | 0.0045 | 0.0062 | >10 | 0.102 | 0.118 | 0.231 | 0.081 | 0.214 | 0.061 | 0.865 | 0.085 | 0.432 | 0.936 | 1.174 | 0.187 | 0.480 | >10 | >10 | >10 | >10 | >10 | >10 |
| KXD05 | 89 | 94 | 23 | 96 | 0.0060 | 0.0046 | 0.0042 | 0.0020 | 0.0034 | 1.7940 | 0.021 | 0.039 | 0.039 | 0.018 | 0.069 | 0.011 | 0.223 | 0.017 | 0.409 | 1.376 | 2.224 | 0.123 | 0.710 | >10 | >10 | >10 | >10 | >10 | >10 |
| KXD06 | 86 | 91 | 12 | 95 | 0.0092 | 0.0071 | 0.0093 | 0.0059 | 0.0067 | >10 | 0.022 | 0.024 | 0.017 | 0.012 | 0.043 | 0.018 | 0.083 | 0.016 | 0.132 | 0.944 | 2.580 | 0.173 | 0.487 | >10 | >10 | >10 | >10 | >10 | >10 |
| KXD07 | -60 | 42 | 98 | 23 | 0.0112 | 0.0072 | >10 | 9.5790 | 8.8310 | >10 | 1.149 | >10 | >10 | >10 | >10 | >10 | >10 | >10 | >10 | >10 | >10 | >10 | >10 | >10 | >10 | >10 | >10 | >10 | >10 |
| KXD08 | 32 | 39 | 97 | 35 | 0.0173 | 0.0202 | 0.0255 | 0.0076 | 0.0106 | 0.0286 | 2.980 | >10 | 1.842 | 1.133 | 1.089 | 1.165 | 0.311 | >10 | >10 | >10 | >10 | 0.324 | >10 | >10 | >10 | >10 | >10 | >10 | >10 |
| KXD09 | 14 | 48 | 98 | 36 | 0.0188 | 0.0197 | 0.0046 | 0.0049 | 0.0032 | 0.0147 | 0.064 | 1.078 | 0.011 | 0.020 | 0.004 | 0.026 | 0.005 | 3.111 | 4.970 | 1.636 | 0.790 | 0.661 | 0.302 | 4.202 | 4.205 | 2.946 | 6.458 | 1.526 | >10 |
| COV2-2196 | 90 | 92 | 43 | 98 | 0.0121 | 0.0102 | >10 | 3.2960 | >10 | >10 | 0.004 | 0.007 | >10 | >10 | >10 | >10 | >10 | 0.159 | >10 | >10 | >10 | >10 | >10 | >10 | >10 | >10 | >10 | >10 | >10 |
| COV2-2130 | 97 | 59 | 99 | 34 | 0.0029 | 0.0069 | 0.0257 | >10 | >10 | >10 | 0.011 | 0.359 | >10 | 0.024 | 0.162 | 0.029 | 0.115 | >10 | >10 | >10 | >10 | >10 | >10 | >10 | >10 | >10 | >10 | >10 | >10 |
| P2C-1F11 | 98 | 88 | 36 | 97 | 0.0039 | 0.4742 | 1.7160 | >10 | >10 | >10 | 0.030 | 0.041 | >10 | 0.567 | >10 | 3.274 | 1.043 | 0.592 | >10 | >10 | >10 | >10 | >10 | >10 | >10 | >10 | >10 | >10 | >10 |
| P5-2H11 | 90 | 86 | 40 | 94 | 0.0114 | 4.9780 | 0.7750 | 4.0070 | 4.3190 | >10 | 0.014 | 0.050 | 0.053 | 0.129 | 0.078 | 0.399 | >10 | 0.055 | >10 | >10 | >10 | >10 | >10 | >10 | >10 | >10 | >10 | >10 | >10 |
| P5S-2B10 | 92 | 81 | 35 | 93 | 0.0059 | 0.2013 | 0.1146 | 0.2262 | 0.2239 | >10 | 0.026 | 0.140 | 0.033 | 0.063 | 0.062 | 0.028 | 0.138 | 0.078 | >10 | 5.073 | >10 | 1.930 | 2.238 | >10 | >10 | >10 | >10 | >10 | >10 |
| BD55-1205 | 93 | 97 | -122 | 94 | 0.0097 | 0.0041 | 0.0037 | 0.0066 | 0.0049 | 0.0105 | 0.025 | 0.035 | 0.069 | 0.018 | 0.033 | 0.017 | 0.018 | 0.013 | 0.012 | 0.022 | 0.036 | 0.010 | 0.015 | 0.019 | 0.015 | 0.035 | 0.022 | 0.040 | >10 |
| BD56-1854 | 96 | 98 | -74 | 96 | 0.0106 | 0.0024 | 0.0041 | 0.0052 | 0.0068 | 0.0057 | 0.033 | 0.016 | 0.009 | 0.005 | 0.010 | 0.006 | 0.006 | 0.005 | 0.017 | 0.014 | 0.012 | 0.008 | 0.008 | 0.037 | 0.018 | 0.041 | 0.065 | 0.216 | >10 |

**Fig 3. Comparison of epitope, binding activity and neutralizing activity among KXD01-09 and reported antibodies.** Epitopes were mapped by competition ELISA. ACE2, COV2-2196, COV2-2130 and P2C-1F11 were labeled with biotin. The percentages of their binding to RBD competed by KXD01-09 and reported antibodies were measured by ELISA. The values are color-coded with darker red indicates more competition. The binding activity against WT and variant RBDs was measured by ELISA. The neutralizing activity was measured by pseudoviruses. The highest antibody concentration used to determine EC50 and IC50 is 10 µg/ml. EC50s and IC50s were calculated by least squares fit. The values of EC50s and IC50s are color-coded with dark blue indicates lower values and dark red indicates higher values. All results are representatives of two independent experiments, in which duplicates are performed.

COV2-2196, COV2-2130 and P2C-1F11 are the prototypes of Tixagevimab, Cilgavimab and Amubarvimab, which were approved for emergency use [1]. In addition to P2C-1F11, P5-2H11, P5S-2B10, BD55-1205 and BD56-1854 are also IGHV3-53/3-66 public antibodies [8,23]. Moreover, P5S-2B10, BD55-1205 and BD56-1854 have broad neutralizing activity across VOCs.

We performed competition ELISA with biotin-labeled ACE2, COV2-2196, COV2-2130 and P2C-1F11 (Fig 3). As expected, KXD01-06 block the binding of RBD with ACE2, COV2-2196 and P2C-1F11, further confirming that they target similar epitope as other IGHV3-53/3-66 public antibodies and neutralize SARS-CoV-2 by ACE2 blocking. Although KXD07-09 compete with COV2-2130, they do not compete with ACE2 like COV2-2130. Therefore, the epitopes of KXD07-09 remain elusive.

We also measured the binding activity of these mAbs against WT and variant RBDs by ELISA (Figs 3 and S2A). KXD01-03, KXD08 and KXD09 maintain binding activity to all tested RBDs. In contrast, KXD04-06 show poor binding to EG.5.1 RBD and KXD07 only exhibits strong binding to WT and BA.2.75 RBDs. Among the published mAbs, BD55-1205 and BD56-1854 show consistent binding to all RBDs, whereas the other mAbs lose binding to variant RBDs with different degrees.

We also tested the neutralizing activity of these mAbs against the same panel of pseudoviruses used for plasma samples (Figs 3 and S2B). Surprisingly, KXD01-06 exhibit broad neutralizing capacity against SARS-CoV-2 viruses. Specifically, KXD01-03 are able to neutralize variants ranging from WT to emerging variants including EG.5, EG.5.1, FL.1.5 and FL.1.5.1, although they show reduced potency against later variants than earlier variants. The neutralizing breadth of KXD04-06 ranges from WT to XBB.1.16, which is consistent with their binding breadth. KXD09 is another bnAb with reduced neutralizing potency against multiple variants including Delta, BF.7, BQ.1, XBB lineages and the emerging variants. In contrast to other KXD antibodies, KXD07 and KXD08 have limited neutralizing breath and potency. Among the published mAbs, BD55-1205 and BD56-1854 show broad and potent neutralizing

antibodies against all SARS-CoV-2 viruses, while the other antibodies have limited neutralizing breadth. Taken together, we identified 6 IGHV3-53/3-66 public antibodies with broad neutralizing capacity against SARS-CoV-2.

## Mapping escape mutations of KXD01-06

To explore the molecular basis for broad neutralization and neutralization escape, we performed deep mutational scanning (DMS) to map escape mutations of KXD01-06 (Figs 4A and S3). We first filtered out RBD mutants losing binding to ACE2 from two independently-constructed mutant libraries based on BA.4/5 RBD. Sequence analysis of yeasts after this round of sorting reveals that the distribution of mutations is rather random (S3A and S3B Fig). Then we sorted RBD mutants with reduced binding to antibodies. In previously reported method [25,26], mutants post second sorting are directly processed to sequencing. Here we performed

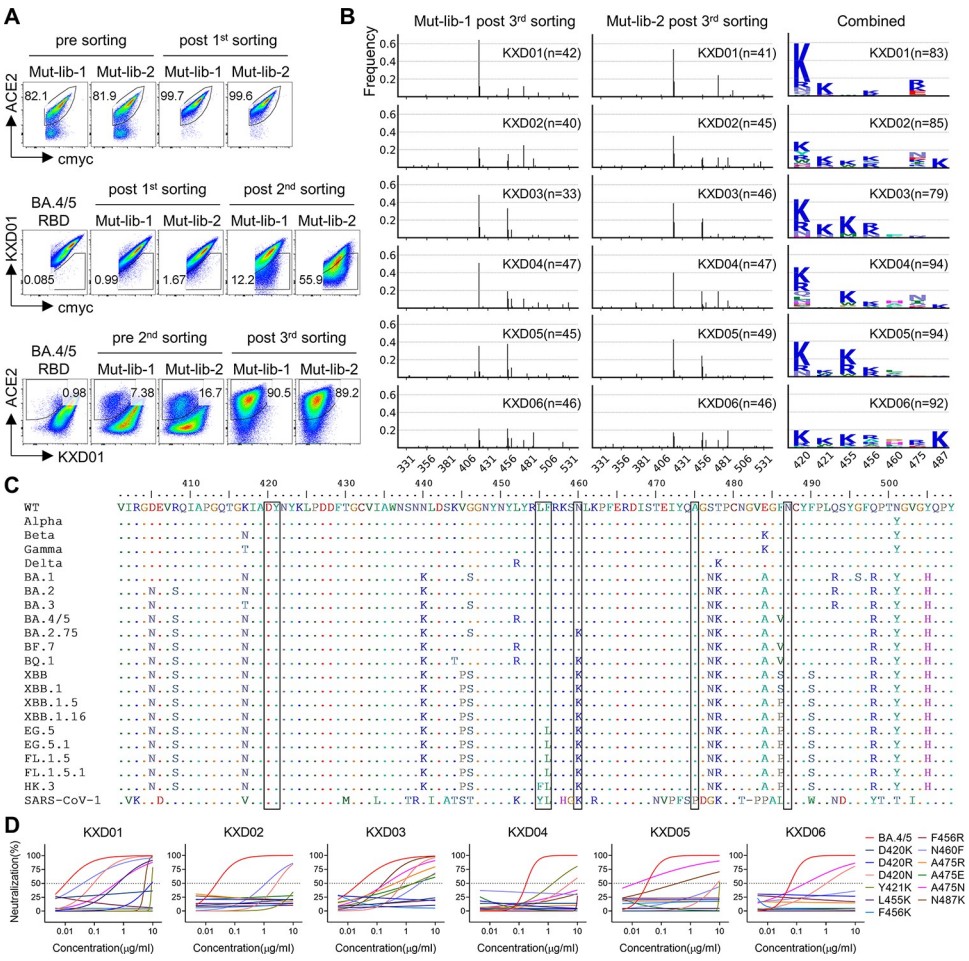

**Fig 4. Mapping of escape mutation.** (A) FACS plots illustrating the process of library screening. In the first sorting, ACE-2-positive mutants were sorted. In the second sorting, mAb-negative mutants were sorted. In the third sorting, ACE2-positive but mAb-negative mutants were sorted. (B) Sequence analysis of mutants post the third sorting. The sequences were aligned with BA.4/5 RBD and then the mutations were identified. The left two columns represent mutation frequency across RBD (331–531). The right column represents mutation profiles on 420, 421, 455, 456, 460, 475 and 487. (C) Mutation profile of representative spike sequences. (D) Neutralizing activity of KXD01-06 against BA.4/5 with escape mutations. Data are represented as non-linear fit curves calculated by least squares fit. For (A-B), two independently constructed libraries are screened as two independent experiments. For (D), the results are representatives of two independent experiments, in which duplicates are performed.

an additional round of sorting to further remove mutants maintaining binding to antibodies or losing binding to ACE2 (Figs 4A and S3C). Sequence analysis of mutants post third sorting shows that escape mutations of KXD01-06 are limited to residues including D420, Y421, L455, F456, N460, A475 and N487. In addition, the escape mutations of each antibody display a different pattern regarding to frequency, implying that there are minor differences in their interactions with RBD (Fig 4B).

We analyzed the variations on the 7 residues among sequences of WT SARS-CoV-2, representative SARS-CoV-2 variants and SARS-CoV-1 (Fig 4C). No mutations are found on D420, Y421 and N487 and only SARS-CoV-1 shows variation on A475. F456L and N460K are common mutations shared by SARS-CoV-2 variants and SARS-CoV-1. Moreover, mutations on L455 are found in SARS-CoV-1 and the emerging HK.3. To track the variations on these residues, we collected spike sequences from 2019 December to 2023 October (S4 Fig). So far, the mutation frequency on D420, Y421, A475 and N487 are still nearly 0. In contrast, N460K appeared in the second half of 2022 and reached 100% quickly. In the past several months, F456L has become a common mutation among the circulating strains along with the emergence of L455F. The accumulation of mutations on spike is consistent with the gradually reduced neutralizing potency of KXD01-06 against emerging variants.

To confirm the effects of mutations on those residues to neutralization evasion, we measured the neutralizing activity of KXD01-06 against BA.4/5 pseudoviruses with escape mutations (Fig 4D). Overall, KXD01-06 are largely or completely escaped by viruses carrying mutations identified by DMS. It is noteworthy that KXD01 maintains relatively potent neutralizing activity against virus with N460F, consistent with above neutralization profile of KXD01 against variants with N460K such as BA.2.75, BQ.1, XBB lineages. Taken together, these results demonstrate that KXD01-06 can be escaped by current and future variants with mutations on residues including D420, Y421, L455, F456, N460, A475 and N487.

## Genetic basis for broad neutralizing activity of KXD01-06

To illuminate the formation of broad neutralizing activity, we move on to characterize the genetic features of KXD01-06 compared to other IGHV3-53/3-66 public antibodies. We collected 1208 published antibodies and analyzed their mutations. As previously reported, IGHV3-53/3-66 public antibodies generally have limited mutations on IGHV3-53/3-66, with 62.1% antibodies have 0–5 mutations and only 8.7% antibodies have more than 10 mutations (Fig 5A). KXD01-06 have more mutation than most IGHV3-53/3-66 antibodies, with 18, 6, 11, 10, 9 and 12 mutations respectively (Mutations probably introduced by PCR primers at the beginning of IGHV3-53/3-66 are not counted) (Fig 5B and 5C). From a study performed by Cao et al. [8], we collected 32 IGHV3-53/3-66 antibodies (BD-bnAbs) with broad neutralizing activity. Similar to KXD01-06, BD-bnAbs also have more mutations than most published IGHV3-53/3-66 antibodies (Fig 5B). To examine whether there are some specific mutations enriched by IGHV3-53/3-66 antibodies with broad neutralizing activity, we compared the mutation profiles of IGHV3-53/3-66 antibodies (Fig 5C). Mutations accumulated by KXD01-06 and BD-bnAbs, such as Y58F, F27I, F27L, T28I, S31R, S35T, S35N and V50L, are also highly frequent among the other IGHV3-53/3-66 antibodies, suggesting that mutations enriched by broadly neutralizing IGHV3-53/3-66 antibodies are rather common instead of specific.

To investigate the function of those mutations, we reverted the IGHV region of KXD01-06 to IGHV3-53/3-66 germline sequences. Then we compared the binding activity of mature antibodies and germline antibodies to WT and variant RBDs (Figs 5D and S5). Although each pair of antibodies bind WT RBD with similar affinities, the germline versions partially or almost completely lose binding to BA.2.75, BA.4/5, BQ.1, XBB.1.5 and EG.5.1 RBDs.

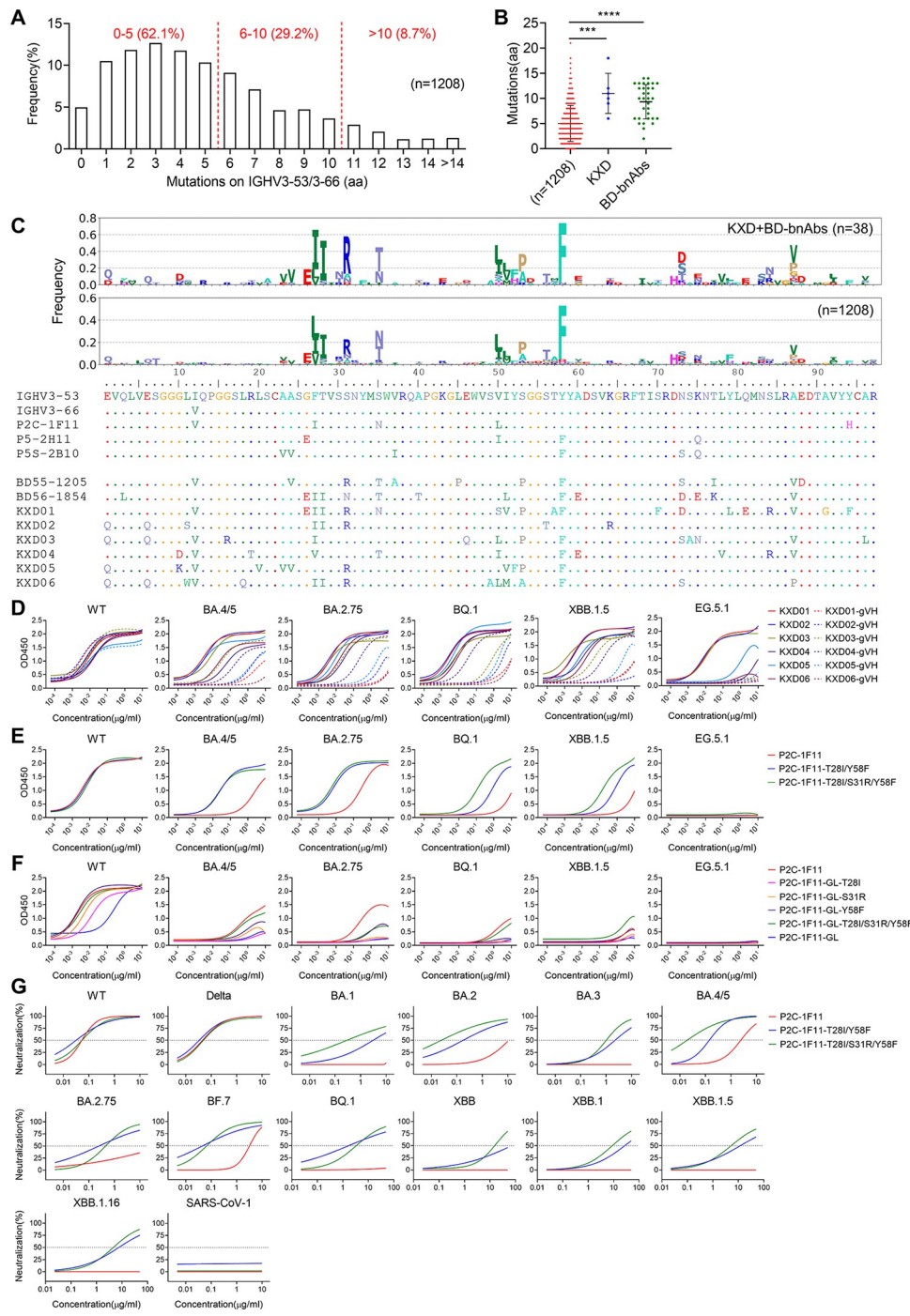

**Fig 5. Effects of somatic hypermutation on neutralization breadth.** (A) Summary of mutations on IGHV3-53/3-66 accumulated by 1208 IGHV3-53/3-66 public antibodies against SARS-CoV-2. (B) Mutation levels of KXD01-06, BD-bnAbs and 1208 reported antibodies. (C) Mutation profiles of IGHV3-53/3-66 antibodies. (D) Binding activity of KXD01-06 and their germline-VH versions measured by ELISA. (E) Binding activity of P2C-1F11 and P2C-1F11 with additional mutations measured by ELISA. (F) Binding activity of P2C-1F11 germline and P2C-1F11 germline with additional mutations measured by ELISA. (G) Neutralizing activity of P2C-1F11 and P2C-1F11 with additional mutations against a panel of pseudoviruses. For BQ.1 and XBB lineages, the starting concentration is 50 μg/ml. For other viruses, the starting concentration is 10 μg/ml. For (D-G), the binding and neutralizing data are represented as non-linear fit curves calculated by least squares fit. The results are representatives of two independent experiments, in which duplicates are performed.

Oppositely, we introduced common mutations including T28I, S31R and Y58F to IGHV region of P2C-1F11 as it lacks these mutations. Consistent with above results, these mutations dramatically increase the binding activity of P2C-1F11 to variant RBDs while not to WT RBD (Figs 5E and S5). Previous studies reported that Y58F can increase the binding of IGHV3-53/ 3-66 antibodies to WT RBD by 10–1000 fold [27,28]. To figure out this inconsistency, we introduced T28I, S31R and Y58F to the germline version of P2C-1F11 (P2C-1F11-GL) individually or in combination (Figs 5F and S5). Consistent with previous findings, these mutations substantially increase the binding of P2C-1F11-GL to WT RBD. However, these mutations have limited or little effects on the binding of P2C-1F11-GL to BA.4/5, BA.2.75, BQ.1, XBB.1.5 and EG.5.1 RBDs, suggesting that more mutations beyond common mutations such as T28I, S31R and Y58F are required to confer P2C-1F11-GL high affinity to variant RBDs. We also analyzed the neutralizing breadth of P2C-1F11 with T28I/Y58F and T28I/S31R/Y58F. Consistent with a prior study [28], these mutations substantially improve the neutralizing activity of P2C-1F11 against Omicron variants, although the IC50s against BQ.1 and XBB lineages are still rather high. Taken together, these results elucidate that the extent of somatic hypermutation is critical for the neutralization breadth of IGHV3-53/3-66 public antibodies.

## Discussion

In this study, we mainly characterized the antibody response in two convalescents recovered from breakthrough infection probably caused by SARS-CoV-2 BA.5 or BF.7. The top 6 neutralizing mAbs (KXD01-06) among a total of 50 spike-specific mAbs cloned from their B cells are encoded by IGHV3-53/3-66, which once again confirms the prevalence of IGHV3-53/3-66 antibodies in neutralizing antibodies against SARS-CoV-2. It is surprising that KXD01-06 have broad neutralizing activity across VOCs including BQ and XBB lineages, and even including EG.5 and FL.1.5 lineages for KXD01-03, whereas most IGHV3-53/3-66 antibodies identified previously have been escaped by VOCs, particularly Omicron variants [4,16–18]. Several groups reported IGHV3-53/3-66 antibodies with broad neutralizing activity against Omicron variants [20–23,28], whereas it remains to be tested whether those antibodies are escaped by BQ and XBB lineages. Recently, Cao et al. comprehensively measured the neutralizing activity of more than 3000 mAbs against a panel of SARS-CoV-2 pseudoviruses including D614G, BA.1, BA.2, BA.2.75, BA.5, BQ.1.1 and XBB. According to this dataset, 32 out of 181 IGHV3-53/3-66 antibodies broadly neutralize all tested viruses, with 3 from SARS convalescents, 2 from WT vaccinees, 4 from WT convalescents, 7 from BA.1 convalescents, 9 from BA.2 convalescents and 7 from BA.5 convalescents [8]. Considering those mAbs were identified by high-throughput single-cell sequencing of hundreds of thousands of B cells, the frequency of broadly neutralizing IGHV3-53/3-66 antibodies in those convalescents and vaccinees are much lower than these two convalescents in this study. The identification of these two convalescents and KXD01-06 suggests that broadly neutralizing IGHV3-53/3-66 antibodies can be highly abundant in antibody response against SARS-CoV-2, although the underlying mechanism remains to be explored.

The development of bnAbs against highly variable pathogens is a key question for immunology and vaccinology. Currently, the development of HIV-1 bnAbs is most studied and a co-evolution model is proposed based on longitudinal analysis of the race between viruses and B cell lineages [29]. According to this model, envelop glycoprotein (Env) from transmitted/ founder (T/F) virus leads the rare B cell precursors of bnAbs to undergo clonal expansion and somatic hypermutation. Subsequently, B cell lineages with desired mutations are selected by Envs from variants of T/F virus for further diversification. After iterative selections, B cell lineages gradually accumulate high levels of somatic hypermutations and eventually mature to

produce bnAbs. Overall, the development of HIV-1 bnAbs is a rare event due to the following roadblocks. First, the precursor B cells of HIV-1 bnAbs are extremely rare in human B cell repertoire [30]. Second, those precursor B cells generally show poor binding activity with most Envs [31,32]. Third, HIV-1 bnAbs have 10–30% mutations and some mutations are intrinsically improbable [33,34]. Compared with HIV-1 bnAbs, there are no such roadblocks in the development of broadly neutralizing IGHV3-53/3-66 antibodies against SARS-CoV-2. As the dominant antibodies targeting RBD, IGHV3-53/3-66 antibodies are prevalent in human B cell repertoire [35,36]. Moreover, IGHV3-53/3-66 germline antibodies already have moderate to high affinities to SARS-CoV-2 [37,38]. In addition, our study shows that the mutation level of broadly neutralizing IGHV3-53/3-66 antibodies is much lower than HIV-1 bnAbs and those mutations are commonly found in IGHV3-53/3-66 antibodies. These findings suggest that IGHV3-53/3-66 antibodies are promising targets for vaccines aiming to elicit bnAbs against SARS-CoV-2.

Regarding to mutations accumulated during antibody affinity maturation, we show that the mutations on IGHV3-53/3-66 substantially enhance the binding activity of KXD01-06 to variant RBDs instead of WT RBD, and thus we speculate that those mutations are selected by variant RBDs during breakthrough infection, which is consistent with co-evolution model of HIV-1 bnAbs. However, IGHV3-53/3-66 antibodies with broad neutralizing activity are also identified in WT vaccinees and WT convalescents as mentioned above, suggesting that variants are not essential for the development of neutralizing breadth across VOCs. Therefore, more studies are required to understand the role of antigen variation in the development of IGHV3-53/3-66 antibodies with broad neutralizing activity.

According to the classification of RBD-specific antibodies [11,39,40], IGHV3-53/3-66 antibodies mainly fall into two groups: RBD class 1 (RBS-A; RBD-2a) and RBD class 2 (RBS-B; RBD-2b). IGHV3-53/3-66 antibodies in RBD class 1 have short CDRH3 and bind to RBM using germline-encoded NY and SGGS motifs in CDRH1 and CDRH2. They only bind to RBD in the up conformation and neutralize SARS-CoV-2 by ACE2 blocking. In contrast, IGHV3-53/3-66 antibodies in RBD class 2 have long CDRH3 and can bind to RBD in both up and down conformation. They mainly contact with the RBD ridge and its nearby regions, and also neutralize SARS-CoV-2 by ACE2 blocking. According to this classification, KXD03, which has a long CDRH3 with 21 amino acids, is supposed to target different epitopes with the other 5 mAbs. Indeed, KXD01-06 share similar escape map with P2C-1F11, which is a representative of RBD class 1 antibody with escape mutations distributed on D420, Y421, L455, F456, N460, P463, Y473, A475 and N487 [26]. On the other hand, although IGHV3-53/3-66 antibodies with broadly neutralizing activity have been widely reported and characterized [8,19–23,28,41–43], they generally use short CDRH3. Here we show that KXD03, which has a long CDRH3, is able to neutralize all tested SARS-CoV-2 viruses and is even more potent to variants with L455F than KXD01 and KXD02. Taken together, these observations suggest that KXD03 represents a unique type of IGHV3-53/3-66 antibodies with long CDRH3, which will be characterized structurally in future.

Regarding to the surveillance of SARS-CoV-2, emerging variants like EG.5, EG.5.1, FL.1.5 and FL.1.5.1 have F456L mutation, which falls into the escape maps of KXD01-06. Although these variants show similar resistance to plasma neutralization as XBB and XBB.1, they partially escape the neutralization of KXD01-03 and completely escape the neutralization of KXD04-06. Moreover, addition of L455F can further increases the neutralization resistance of EG.5.1 to KXD01-03. Consistent with studies published recently [9,10], these results demonstrate the evolution potential of SARS-CoV-2 and highlight the need to monitor current variants with those mutations.

We acknowledge that there are several potential limitations of this study. First, we only identified 50 spike-specific mAbs, which may be not enough to represent the antibody response in the two individuals. Second, we did not collect blood samples before the breakthrough infection. So we are not able to compare IGHV3-53/3-66 antibodies generated before and after the breakthrough infection, which is helpful to understand the development of broadly neutralizing IGHV3-53/3-66 antibodies. Finally, the profiles of neutralization and escape mutations suggests that there are some minor differences in the interactions of KXD01-06 with spike. In future studies, structure analysis can be performed to further elucidate these differences.

The rapid emergence of SARS-CoV-2 VOCs highlights the urgent need to develop vaccines and antibody therapeutics with broad protection against current and future SARS-CoV-2 variants. This study demonstrates that IGHV3-53/3-66 public antibodies have enormous potential to develop broad and potent neutralizing activity through antibody affinity maturation, which provides rationale for the development of novel vaccines and antibody therapeutics based on this class of antibodies.

## Materials and methods

### Ethics statement

This study was approved by the Ethics Committee of Hefei Institutes of Physical Science, Chinese Academy of Sciences (Approval Number: YXLL-2023-47). All donors provided written informed consent for collection of information, analysis of plasma and PBMCs, and publication of data generated from their samples.

### Human samples

Peripheral blood samples were collected from 11 donors (Table 1). Plasma and peripheral blood mononuclear cells (PBMCs) were separated from blood by Ficoll density gradient centrifugation.

### Cell lines

HEK-293T cells expressing human ACE2 (HEK-293T-hACE2) were kindly provided by Prof. Ji Wang at Sun Yat-Sen University. HEK-293T cells were from ATCC. HEK-293T and HEK-293T-hACE2 cells were cultured in DMEM with 10% FBS and 1% penicillin/streptomycin (pen/strep). FreeStyle 293 F cells (Thermo Fisher Scientific) were cultured in SMM 293-TII Expression Medium (Sino Biological Inc., M293TII). All cells were maintained in a 37°C incubator at 5% $CO_2$.

### Protein expression and purification

The genes encoding the extracellular domain of SARS-CoV-2 spike including WT, Omicron BA.4/5, Omicron XBB.1.5 were constructed with a foldon trimerization motif and tandem strep tags at C-terminal. The genes encoding WT, BA.2.75, BA.4/5, BQ.1, XBB.1.5 and EG.5.1 RBDs were constructed with a strep tag at C-terminal. Spike trimer and RBD were expressed in the FreeStyle 293 F cells and purified by BeaverBeads Strep-Tactin (BEAVER, 70808–250).

### ELISA

RBD or spike were coated onto 96-well ELISA plates (100 ng/well) and incubated at 4°C overnight. After blocking with PBS containing 10% fetal bovine serum (FBS), 3-fold serially diluted plasma (starting at 1:100), mAbs (starting at 10 µg/ml) or HEK293T supernatant were added

to the wells and incubated at 37˚C for 1 hr. HRP–conjugated goat anti-human antibodies (Zen-bio, 550004; 1:5000 dilution) were added to the wells and incubated at 37˚C for 1 hr. TMB substrate (Sangon Biotech, E661007-0100) was added to the wells and incubated at room temperature for 5 mins. The reaction was stopped by TMB Stop Solution (Sangon Biotech, E661006-0500) and absorbance at 450 nm was measured.

## Pseudovirus neutralization assay

To generate pseudoviruses, HEK-293T cells were transfected with psPAX2, pLenti-luciferase and spike-encoding plasmids using polyetherimide (PEI). Supernatant with pseudovirus was collected 48 hrs after transfection. 3-fold serially diluted plasma (starting at 1:20), HEK293T supernatant (starting at 1:4.5), or mAbs (starting at 10 μg/ml) were mixed with pseudoviruses at 37˚C for 1 hr. HEK293T-hACE2 cells ($1.5\times10^4$ per well) were added into the mixture and incubated at 37˚C for 48 hrs. Cells were lysed to measure luciferase activity (Bright-Lite Luciferase Assay System, DD1204-02 Vazyme Biotech Co., Ltd.). The percent of neutralization was determined by comparing with the virus control. The plasmids encoding spike of WT SARS-CoV-2, Delta, BA.1, BA.2, BA.3, BA.4/5, BF.7, BQ.1, XBB, XBB.1, XBB.1.5, XBB.1.16 and SARS-CoV-1 were kindly provided by Prof. Linqi Zhang at Tsinghua University. The plasmids encoding spike of BA.2.75 was kindly provided by Prof. Zezhong Liu at Fudan University. The plasmids encoding EG.5, EG.5.1, FL.1.5, FL.1.5.1, HK.3 and BA.4/5 spike with escape mutations were generated by PCR with primers containing mutations.

## Spike-specific single B cell sorting and mAb cloning

PBMCs from Donor 1 and 2 were incubated with 200 nM SARS-CoV-2 spike for 30 min at 4˚C. After wash, they were stained with cell-surface antibodies: CD3-BV510 (BioLegend, 317331), CD19-PE/Cy7 (BioLegend, 302215), CD27-APC (BioLegend, 356409), CD38-APC/Cy7 (BioLegend, 356615), human IgM-AF700 (BioLegend, 314537), human IgD-perCP/Cy5.5 (BioLegend, 348207), anti-His-FITC (Proteintech, CL488-66005), anti-FLAG-PE (BioLegend, 637309) and DAPI. The stained cells were washed with FACS buffer (PBS containing 2% FBS) and resuspended in 500 μl FACS buffer. Spike-specific single B cells were gated as DAPI⁻CD3⁻CD19⁺CD27⁺IgD⁻His⁺FLAG⁺ and sorted into 96-well PCR plates containing 4 μl of lysis buffer (0.5×PBS, 0.1 M DTT and RNase inhibitor) per well. After reverse transcription reaction, the heavy and light chain variable regions were amplified by nested PCR and cloned into expression vectors to produce full IgG1 antibodies [44]. The paired heavy- and light-chain were co-transfected into HEK-293T cells or FreeStyle 293 F cells. Antibodies were purified with Protein A magnetic beads (GenScript, L00273). Expression vectors for COV2-2196, COV2-2130, P2C-1F11, P5-2H11, P5S-2B10 were kingly provided by Prof. Linqi Zhang at Tsinghua University. The variable regions of BD55-1205 and BD56-1854 were synthesized and cloned into expression vectors.

## Competition ELISA

WT RBD was coated onto 96-well ELISA plates (100 ng/well) and incubated at 4˚C overnight. After blocking with PBS containing 10% FBS, biotinylated ACE2, COV2-2196, COV2-2130 and P2C-1F11 were mixed with competing mAbs at 1:50 molar ratio. The mixtures were added to the wells and incubated at 37˚C for 1 hr. Streptavidin-HRP (GenScript, M00091; 1:5000 dilution) was added to the wells and incubated at 37˚C for 1 hr. TMB substrate (Sangon Biotech, E661007-0100) was added to the wells and incubated at room temperature for 5–10 mins. The reaction was stopped by TMB Stop Solution (Sangon Biotech, E661006-0500) and

absorbance at 450 nm were measured. The percentages of signal decrease caused by competing mAbs were calculated.

## Deep mutational scanning

Yeast libraries displaying BA.4/5 RBD mutants were kindly provided by Prof. Yunlong Cao at Peking University. Three rounds of FACS were performed to enrich RBD mutants losing binding to mAb but maintaining binding to ACE2. In the first round, yeasts were stained with ACE2 and ACE2-positive yeasts were sorted and expanded. In the second round, yeasts were stained with mAbs and mAb-negative yeasts were sorted and expanded. In the third round, yeasts were stained with ACE2 and mAbs simultaneously. ACE2-postive but mAb-negative yeasts were sorted. With the yeasts post the third sorting as template, PCR was performed to amplify RBD fragment from the plasmid. The PCR products were cloned to T vector and sequenced by Sanger-sequencing. The sequences are aligned with reference (BA.4/5) and mutations are identified. Then the frequency of mutations on each amino acid is analyzed.

## Data analysis

Sequences of antibody heavy and light chain variable regions from single B cells were analyzed with IgBlast (https://www.ncbi.nlm.nih.gov/igblast/). Sequence alignment was performed either by MEGA or by the Muscle v5 algorithm. Sequence logos displaying mutation profiles were created with the Pandas, Bio, Matplotlib, Seaborn, and Logomaker packages in Python 3.8.16 environment. FACS data were analyzed with Flowjo. ELISA and neutralizing data were analyzed and plotted with Graphpad Prism 8.

## Supporting information

**S1 Fig. Isolation and characterization of mAbs from Donor 1 and 2.** (A) FACS plots representing gating strategies for spike-specific single B cell sorting. (B) Summary of 22 spike-negative mAbs.
(TIF)

**S2 Fig. Measurement of binding activity and neutralizing activity of KXD01-09 and reported antibodies.** (A) Binding activity against WT and variant RBDs measured by ELISA. (B) Neutralizing activity against a panel of pseudovirus. The data are represented as non-linear fit curves calculated by least squares fit. All results are representatives of two independent experiments, in which duplicates are performed.
(TIF)

**S3 Fig. Mapping of escape mutation.** (A) Sequence analysis of mutants post the first sorting. (B) Mutation profile of yeasts post the first sorting. (C) FACS plots of the second and the third sorting of two RBD mutant libraries by KXD01-06.
(TIF)

**S4 Fig. Mutation profile of SARS-CoV-2 spike sequences.**
(TIF)

**S5 Fig. Fold change of EC50.** EC50s against RBDs were measured by ELISA with starting concentration of 10 μg/ml. For antibodies with poor binding activity to RBDs, EC50s were taken as >10 μg/ml. To calculate the fold change of EC50, higher values were used as numerator and lower values were used as denominator. If both EC50s were >10 μg/ml, then "N/A" is recorded, indicating not applicable. If the numerator is >10 μg/ml, then ">" is used in the fold

change. The values in the table were color-coded.
(TIF)

**S1 Table. Spike sequences of SARS-CoV-2 variants and SARS-CoV-1.**
(XLSX)

**S1 Data. Excel spreadsheet containing numerical data for figures.**
(XLSX)

## Acknowledgments

The authors thank all donors for providing the blood samples. The authors thank Prof. Linqi Zhang at Tsinghua University, Prof. Ji Wang at Sun Yat-Sen University, Prof. Yunlong Cao at Peking University and Prof. Zezhong Liu at Fudan University for providing key materials for this study. The authors thank Zexuan Zheng at University of Science and Technology of China for help with yeast sorting.

## Author Contributions

**Conceptualization:** Xixian Chen, Liwei Jiang, Teng Zuo.

**Data curation:** Teng Zuo.

**Formal analysis:** Xixian Chen, Zuowei Wang, Teng Zuo.

**Funding acquisition:** Ling Li, Liwei Jiang, Teng Zuo.

**Investigation:** Ling Li, Xixian Chen, Zuowei Wang, Yunjian Li, Chen Wang, Liwei Jiang.

**Methodology:** Liwei Jiang, Teng Zuo.

**Project administration:** Liwei Jiang.

**Software:** Yunjian Li.

**Supervision:** Liwei Jiang, Teng Zuo.

**Validation:** Ling Li, Xixian Chen, Zuowei Wang, Yunjian Li, Liwei Jiang.

**Visualization:** Yunjian Li, Teng Zuo.

**Writing – original draft:** Teng Zuo.

**Writing – review & editing:** Ling Li, Xixian Chen, Zuowei Wang, Yunjian Li, Liwei Jiang, Teng Zuo.

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
