## [Decision Letter · Decision Letter 0]

10 Sep 2023

Dear Dr Zuo,

Thank you very much for submitting your manuscript "Breakthrough infection elicits hypermutated IGHV3-53/3-66 public antibodies with broad and potent neutralizing activity against SARS-CoV-2 variants including BQ and XBB lineages" for consideration at PLOS Pathogens. As with all papers reviewed by the journal, your manuscript was reviewed by members of the editorial board and by several independent reviewers. In light of the reviews (below this email), we would like to invite the resubmission of a significantly-revised version that takes into account the reviewers' comments.

Thanks for submitting the paper. We have received two expert reviews that suggest some important revisions. Please submit a revised version that responds to all of these comments, including especially discussing new mutations such as at site 456 in the RBD that are becoming common in current SARS-CoV-2 variants.

We cannot make any decision about publication until we have seen the revised manuscript and your response to the reviewers' comments. Your revised manuscript is also likely to be sent to reviewers for further evaluation.

Sincerely,

Jesse D Bloom, Ph.D.

Guest Editor

PLOS Pathogens

Alexander Gorbalenya

Section Editor

PLOS Pathogens

Kasturi Haldar

Editor-in-Chief

PLOS Pathogens

orcid.org/0000-0001-5065-158X

Michael Malim

Editor-in-Chief

PLOS Pathogens

orcid.org/0000-0002-7699-2064

Thanks for submitting the paper. We have received two expert reviews that suggest some important revisions. Please submit a revised version that responds to all of these comments, including especially discussing new mutations such as at site 456 in the RBD that are becoming common in current SARS-CoV-2 variants.

Reviewer's Responses to Questions

**Part I - Summary**

Reviewer #1: IGHV3-53/3-66 genes encode a major public class of neutralizing antibodies targeting the receptor binding site of the SARS-CoV-2 spike protein. Although a vast majority of IGHV3-53/3-66 antibodies are evaded by SARS-CoV-2 variants, exceptions have been discovered - some IGHV3-53/3-66 antibodies exhibited broad reactivity and neutralizing activity against the variants. In this manuscript, the authors identified six IGHV3-53/3-66 antibodies that can neutralize a broad range of variants from WT to XBB.1.16. However, while breakthrough infections eliciting IGHV3-53/3-66 antibodies with great breadth and potency have been widely reported and discussed, the authors should highlight the new observations and added knowledge from this study.

Reviewer #2: The authors identified a panel of broadly neutralizing antibodies against multiple SARS-CoV-2 variants including XBB lineages, named KXD01-06, which belong to the IGHV3-53/3-66 public antibody family and target the ACE2-binding sites on the virus RBD. Although these antibodies exhibit attractive neutralization breadth in the manuscript’s characterization, some additional validation experiments are necessary and some concerns should be addressed before my recommendation for publication.

**Part II – Major Issues: Key Experiments Required for Acceptance**

Reviewer #1: 1. Deep mutational scanning experiments here have demonstrated that some mutations on residues 455, 456, and 460 can lead to KXD01-06 antibody escape (Figure 4). Variants EG.5.1 and FL.1.5.1 are rapidly rising and contain mutations at these positions. Can the authors investigate the potential impact of these emerging variants on these antibodies? Based on these considerations, relevant sections of the manuscript may need to be modified, for example:

Line 24: “conserved sites on RBD including D420, Y421, L455, F456, A475 and N487”

Lines 48-49: “the emerging BQ and XBB sublineages, display most striking neutralization evasion” – the new variants may be even more resistant to antibody neutralization (DOI: 10.1101/2023.08.21.553968).

Lines 187-189: “KXD01-06 target highly conserved epitopes and they can be escaped by future variants with mutations on residues including D420, Y421, L455, F456, N460, A475 and N487”

Line 173: “mutations are only found in SARS-CoV-1 on L455, F456”

2. Many previous studies have identified IGHV3-53/3-66 bnAbs (some also neutralize XBB), and have extensively discussed potential reasons for their breadth. e.g. DOIs 10.1186/s12929-023-00955-x, 10.1038/s41421-022-00449-4, 10.1101/2022.01.03.474825, 10.1126/sciimmunol.abo3425. It is important to show new findings of this study.

3. A major claim of this study was that the common somatic mutations VH T28I, S31R and Y58F, when introduced to an IGHV3-53/3-66 antibody, significantly enhanced its affinity and neutralization. A prior study already reported a similar finding, where similar mutations were demonstrated to confer binding and breadth to this class of antibodies (DOI: 10.1101/2022.01.03.474825) but not cited or discussed here. Additionally, it is interesting why P2C-1F11, adding the three mutations, still did not neutralize XBB.

Reviewer #2: Major points

1. F456L is a mutation convergently observed in multiple XBB*+486P lineages, such as XBB.1.5.10, EG.5, and FL.1.5. As F456 is exactly on the epitope recognized by the identified IGHV3-53/3-66 neutralizing antibodies, it is crucial to test the activity of KXD01-06 against these XBB variants with F456L, at least using pseudovirus.

2. Line 175. Considering the evolution and circulation of SARS-CoV-2 VOCs, it is better to sample more recently uploaded sequences, which is much more reasonable, given that previous sequences are not likely to circulate again due to their weak capability of escaping neutralizing antibodies. If you use recent sequences for the analysis, you should find that nearly all sequences carry N460K and F456L is a frequent mutation.

3. SHM of IGHV3-53/3-66 is analyzed in the study. However, only the common mutations in published IGHV3-53/3-66 antibodies are summarized in Figure 5C. Can you identify some specific SHMs that exhibit enrichment in the broadly neutralizing antibodies but not the escaped antibodies?

4. It is mentioned that Sanger sequencing is used to identify escape mutations in the DMS experiments. However, because the frequencies of yeast cells carrying different variants could be quite different in the library, it is usually necessary to sequence both the mAb-negative yeasts and the pre-antibody-selection yeasts for normalization. This step is not mentioned in the methods. Can you give a detailed methods for the analysis the DMS datasets and the illustration of Figure 4B? The heights of “K” and “R” seem too high in Figure 4B and there may be some artifacts.

**Part III – Minor Issues: Editorial and Data Presentation Modifications**

Reviewer #1: 1. Lines 209-212: “common mutations including T28I, S31R and Y58F … dramatically increase the binding activity of P2C-1F11 to variant RBDs while not to WT RBD”. This observation contrasts with previous findings where Y58F increased binding of this class of antibodies to WT RBD by 10-1000 fold (DOIs: 10.1038/s41467-021-24123-7, 10.1101/2022.01.03.474825). The authors may want to discuss this.

2. What is the antibody sequence numbering system used in this study? Kabat, IMGT or Chothia? For example, in line 122 “short CDRH3 ranging from 10 to 12 amino acids or long CDRH3 of 21 amino acids”. Different numbering systems can result in varying definitions of CDR lengths. Y58F is used throughout the manuscript but “Y57F” is shown in line 201.

3. Line 267: “IGHV3-35” should be replaced by “IGHV3-53”

4. Line 273: “(ref)”

5. Line 294: “KCD” should be replaced by “KXD”?

Reviewer #2: Minor points

1. Line 72. It will be much clearer for the readers to specify the RBD mutations in the VOCs that are responsible for the escape of IGHV3-53/3-66 antibodies (mainly K417N and N460K).

2. Line 129. The names of the two prototype NAbs of Evusheld should be COV2-2196/2130 instead of CoV-2196/2130. This should be corrected throughout the manuscript to avoid confusion.

3. Line 201. The authors seem to use an uncommon numbering of antibody chains. For example, a frequent SHM on IGHV3-53/3-66 is previously referred to as Y58F, but here Y57F is used (https://www.nature.com/articles/s41467-021-24123-7;
https://www.science.org/doi/full/10.1126/science.abd2321). Please check the numbering.

PLOS authors have the option to publish the peer review history of their article (what does this mean?). If published, this will include your full peer review and any attached files.

Reviewer #1: No

Reviewer #2: No
---

## [Decision Letter · Decision Letter 1]

16 Nov 2023

Dear Dr Zuo,

Thank you very much for submitting your manuscript "Breakthrough infection elicits dominant IGHV3-53/3-66 public antibodies with broad and potent neutralizing activity against SARS-CoV-2 variants including the emerging EG.5 lineages" for consideration at PLOS Pathogens. As with all papers reviewed by the journal, your manuscript was reviewed by members of the editorial board and by several independent reviewers. The reviewers appreciated the attention to an important topic. Based on the reviews, we are likely to accept this manuscript for publication, providing that you modify the manuscript according to the review recommendations.

Thanks for submitting your revised manuscript. The reviewers are generally supportive. They note (and the handling Editor agrees) that these antibodies may soon be partially escaped by new mutations at sites like 455, 456, and 475 that are present in some of the newest variants. Nonetheless, this remains an interesting study. We just ask that you address the minor comments provided by the reviewers and provide a tracked changes manuscript in the resubmission. If you do that, we should be able to editorially review it and accept it provided you address the minor comments.

Sincerely,

Jesse D Bloom, Ph.D.

Guest Editor

PLOS Pathogens

Alexander Gorbalenya

Section Editor

PLOS Pathogens

Kasturi Haldar

Editor-in-Chief

PLOS Pathogens

orcid.org/0000-0001-5065-158X

Michael Malim

Editor-in-Chief

PLOS Pathogens

orcid.org/0000-0002-7699-2064

Thanks for submitting your revised manuscript. The reviewers are generally supportive. They note (and I agree) that these antibodies may soon be partially escaped by new mutations at sites like 455, 456, and 475 that are present in some of the newest variants. Nonetheless, this remains an interesting study. I just ask that you address the minor comments provided by the reviewers and provide a tracked changes manuscript in the resubmission. If you do that, I should be able to editorially review it and accept it provided you address the minor comments.

Reviewer Comments (if any, and for reference):

Reviewer's Responses to Questions

**Part I - Summary**

Reviewer #1: The authors have addressed most of my comments. I still have two questions.

1. In the revised manuscript, the authors demonstrated that the neutralization by IGHV3-53/3-66 antibodies was reduced or evaded by variants containing F456L and/or L455F (Figure 3). Nonetheless, their plasma exhibited a comparable level of neutralization across variants with and without F456L/L455F mutations (Figure 1 and Response “… these variants show similar resistance to plasma neutralization as XBB and XBB.1 …”). This suggests that antibodies other than IGHV3-53/3-66 could contribute significantly to the plasma neutralization. Consequently, the revised emphasis on the “dominance of IGHV3-53/3-66 antibodies” may not be precise.

2. The authors have included a discussion in the revised manuscript (Lines 314-320) that states:

“On the other hand, although IGHV3-53/3-66 antibodies with broadly neutralizing activity have been widely reported and characterized (8, 19-23, 28, 41-43), few of them are from RBD class 2. Here we show that KXD03 is able to neutralize all tested SARS-CoV-2 viruses and is even more potent to variants with L455F than KXD01 and KXD02. Taken together, these observations suggest that KXD03 represents a unique type of IGHV3-53/3-66 antibodies with long CDRH3, which will be characterized structurally in the future.”

However, IGHV3-53/3-66 antibodies with a long CDR H3 are not novel. For instance, C144 (DOI: 10.1038/s41586-020-2852-1) and COVA2-39 (DOI: 10.1016/j.celrep.2020.108274) have long CDR H3s.

Moreover, while most IGHV3-53/3-66 antibodies with long CDRH3 target class 2 epitope, some bind to the class 1 epitope. For example, the IGHV3-66 antibody BG1-22 has a 19-AA long CDR H3 but is a class 1 antibody (10.1016/j.cell.2021.04.032). In fact, the definition of class 1-4 antibodies is largely based on structural, or at least precise epitope information (10.1038/s41586-020-2852-1). Classifying RBD antibodies without detailed data is challenging.

Reviewer #2: The authors have provided convincing explanation and satisfactory additional experiments and analyses to address my concerns. Although the highlighted mAbs exhibited reduced neutralization activity against latest XBB subvariants with F456L mutation, which may dampen the significance of this study, the analyses and data shown in the study are basically sound and meaningful, helping understand the maturation of IGHV3-53/3-66 public antibodies against SARS-CoV-2 and variants. The long CDR-H3, and the role it played in the neutralization breadth of KXD03 and other IGHV3-53/3-66 mAbs, is also of interest and importance. However, I have the following concerns and suggestions about the modified manuscript.

**Part II – Major Issues: Key Experiments Required for Acceptance**

Reviewer #1: (No Response)

Reviewer #2: (No Response)

**Part III – Minor Issues: Editorial and Data Presentation Modifications**

Reviewer #1: (No Response)

Reviewer #2: 1.EG.5, EG.5.1, and FL.1.5 should have exactly the same RBD sequences, but as shown in Figure 3, KXD01-03 exhibit distinct neutralization against them. Specifically, FL.1.5 seems more resistant to the mAbs than EG.5.1. KXD03 exhibits much lower neutralization against EG.5 (0.764 ug/mL) than EG.5.1 (0.097 ug/mL). I recommend repeating the neutralization assays to confirm the results, if the authors did not perform the assays in replicates, or is it possible to explain them?

2.The authors provided the sequencing results of pre-selection DMS library in Fig. S3A at residue level. Please show the frequencies of each mutation in the pre-selection library, in order to check whether the K/R mutants on residues 420, 421, 456, 475 etc. are originally abundant in the library. If so, it will explain why there are exceptional enrichment of K/R in the DMS results of mAbs, which may be artifacts. Is it possible to normalize the results by the frequency of each mutation in the library before selection?

3.The Spike sequences or mutations of variants involved in this study should be provided as a Supplementary table or figure.

PLOS authors have the option to publish the peer review history of their article (what does this mean?). If published, this will include your full peer review and any attached files.

Reviewer #1: No

Reviewer #2: No

Figure Files:

Data Requirements:

Reproducibility:

References:

---

## [Editor Report · Decision Letter 2]

25 Nov 2023

Dear Dr Zuo,

We are pleased to inform you that your manuscript 'Breakthrough infection elicits hypermutated IGHV3-53/3-66 public antibodies with broad and potent neutralizing activity against SARS-CoV-2 variants including the emerging EG.5 lineages' has been provisionally accepted for publication in PLOS Pathogens.

Best regards,

Jesse D Bloom, Ph.D.

Guest Editor

PLOS Pathogens

Alexander Gorbalenya

Section Editor

PLOS Pathogens

Kasturi Haldar

Editor-in-Chief

PLOS Pathogens

orcid.org/0000-0001-5065-158X

Michael Malim

Editor-in-Chief

PLOS Pathogens

orcid.org/0000-0002-7699-2064

Editor:

Thank you for addressing all of the remaining concerns in the second revision. I am recommending the manuscript for acceptance.
---

## [Editor Report · Acceptance letter]

29 Nov 2023

Dear Dr Zuo,

We are delighted to inform you that your manuscript, "Breakthrough infection elicits hypermutated IGHV3-53/3-66 public antibodies with broad and potent neutralizing activity against SARS-CoV-2 variants including the emerging EG.5 lineages," has been formally accepted for publication in PLOS Pathogens.

Best regards,

Kasturi Haldar

Editor-in-Chief

PLOS Pathogens

orcid.org/0000-0001-5065-158X

Michael Malim

Editor-in-Chief

PLOS Pathogens

orcid.org/0000-0002-7699-2064